# Muscular Fitness Improves during the First Year of Academy Studies among Fighter Pilot Cadets

**DOI:** 10.3390/ijerph17249168

**Published:** 2020-12-08

**Authors:** Tuomas Honkanen, Harri Rintala, Jani P. Vaara, Heikki Kyröläinen

**Affiliations:** 1Aeromedical Centre, Centre for Military Medicine, The Finnish Defense Forces, 00301 Helsinki, Finland; 2Faculty of Sport and Health Sciences, University of Jyväskylä, 40014 Jyväskylä, Finland; harri.rintala@jyu.fi; 3The Department of Leadership and Military Pedagogy, National Defence University, 00860 Helsinki, Finland; jani.vaara@mil.fi (J.P.V.); heikki.kyrolainen@jyu.fi (H.K.); 4Neuromuscular Research Center, Faculty of Sport and Health Sciences, University of Jyväskylä, 40014 Jyväskylä, Finland

**Keywords:** muscular fitness, fighter pilot, military aviation

## Abstract

*Background*: An adequate level of muscular fitness is related to occupational performance in military personnel, including pilots flying high performance aircraft. The aim of this study was to describe the baseline level and the change in muscular fitness between the first and the second years of the Air Force Academy among fighter pilot cadets. *Methods*: The muscular strength and endurance test results of 182 male fighter pilot cadets were analyzed during their first year in the Air Force Academy and one year after. Maximal isometric strength tests included trunk flexion, trunk extension and bilateral leg extension tests, whereas muscle endurance was measured with modified a sit-up test and seated alternative dumbbell press. *Results*: The maximal isometric bilateral strength of the leg extensor muscles increased from 220 ± 42 to 232 ± 42 kg. The maximal isometric trunk extension strength increased from 117 ± 21 to 120 ± 19 kg and trunk flexion from 82 ± 16 to 86 ± 17 kg. Muscle endurance increased from 68 ± 13 to 75 ± 15 repetitions/min in seated dumbbell press and from 47 ± 12 to 51 ± 13 repetitions/min in sit-up test. *Conclusions*: Both maximal strength and muscular endurance improved among fighter pilot cadets, which indicates that occupational performance is well maintained or improved from the perspective of physical fitness during the early phase of academy studies. Education in the Air Force Academy, including physical education, seems beneficial in improving muscular fitness among military pilots.

## 1. Introduction

Physical fitness is a vital component for performing different military tasks including flying high performance aircraft (HPA). Sufficient levels of muscular strength and endurance are both important due to potential physical demands during high Gz (headwards acceleration resulting in a downward force) exposure and anti G-straining maneuvers [1]. Moreover, there is evidence that resistance training improves human tolerance to air combat maneuvering and Gz in terms of prevention of the G-induced loss of consciousness [2]. It seems that the more technical nature of today’s aircrafts has not reduced the physical requirements of military aviators [3]. Previous studies have observed that exposure to Gz force causes fatigue, especially in the neck and low back muscles [4]. In addition, physical activity has an influence on several cognitive tasks [5] and, therefore, it can be seen as a significant contributive factor to operative readiness and flight safety not only to a HPA pilot but also to entire operative organizations. 

It is a common view that the level of physical fitness reflects health status and performance capability in various activities. Particularly, weak trunk muscles appear to be associated with persistent low back pain (LBP), while good isometric endurance of the back muscles seems to prevent the occurrence of LBP among the general working-age population [6]. Respectively, adequate isometric back endurance seems to have a protective role in LBP among military pilots [7], while HPA pilots with physically active sports backgrounds are less likely to be limited to fly due to spinal disorders [8].

Despite the physically demanding high acceleration work environment, no previous studies have investigated effects of physical training during the Air Force Academy (AFA) on maximal strength and muscle endurance among HPA pilots. We were able to find three studies [9,10,11] investigating changes in physical fitness among army or air force cadets with longitudinal designs. Among these studies, Harwood et al. [11] reported improved fitness, strength and muscular endurance during the Commissioning Course in British Army Cadets. Daniels et al. [10], who investigated changes in maximal isometric strength, found significant changes only in upper torso strength in a one-year follow-up among US Army Cadets. A study by Aandstadt et al. [9] was the only study conducted on Air Force cadets. However, their study [9] only investigated the change in aerobic fitness (VO_2_max), not in maximal muscle strength or endurance.

The primary purpose of the present study was to investigate muscular fitness, particularly maximal muscle strength and endurance and the basic somatic characteristics among military pilot cadets between the first and second year of their studies in the Air Force Academy (AFA) in the Finnish Air Force (FINAF).

## 2. Materials and Methods

The study subjects (*n* = 182) were AFA male cadets from 10 different cadet courses between the years of 2007 and 2016. Their mean (±SD) age was 20.6 (±0.6) years at baseline and their mean body mass, height and body mass index (BMI) were 76.4 (±7.2) kg, 180.0 (±5.1) cm and 23.6 (±1.8) kg/m^2^, respectively, at the time of the initial tests. The tests were conducted during the AFA’s physical education course (PEC) for military pilot cadets. Only cadets who were in fighter pilot training were included and thus other FINAF cadets were not included. The average number of participants was 19 per year, ranging from 15 (2007) to 23 (2008) cadets per year. A total of nine cadets could not participate on the second PEC due to common injuries or illnesses i.e., flu-like symptoms. The Ethical Committee of National Defence University in Finland has approved this study (identification code: CK11847).

### 2.1. Study Design

This study was a longitudinal study with one-year follow-up. The baseline tests were conducted during the first year of AFA studies and the follow-up tests a year after. The testing was conducted in PEC, which is a one-week course including not only physical fitness tests but also occupational physical education with individually tailored exercise programs. The testing protocol included three maximal isometric muscle strength tests and two dynamic muscle endurance tests. 

### 2.2. Maximal Isometric Muscle Strength Tests

Prior to all muscle strength tests, the cadets performed a standardized 20 min warm-up. This included light jogging for the first five minutes followed by core and mobility exercises guided by a physiotherapist. The tests were carefully introduced to the subjects and in all tests, verbal encouragement was given to each subject.

Maximal isometric trunk flexion and extension were performed in the standing position, face towards the aperture and the wall, while the flexion test was done in the same aperture, standing, but in the opposite direction (face away from the wall). There is detailed description of the procedure in our previous study [12]. The measurement was recorded by an isometric electromechanical strain-gauge dynamometer (Digitest Ltd., Oulu, Finland). The hips were fixed at the level of the anterior superior iliac spine. The strap was tightened around the shoulders just below the axillary region and horizontally connected to the dynamometer by a steel chain. Two trials were performed for each subject and the better result was selected for further analysis. The duration of maximal pull against the strap was held for 3–5 s and performed twice with 30–60 s rest between the trials.

Maximal isometric bilateral strength of the leg extensor muscles was measured on an electromechanical dynamometer. The subject was positioned sitting on the bench with his back firmly fixed into the backrest and hands on the handles. The subjects placed their feet on the resistance stand at the base of the sledge. The knee angle was set to 90 degrees using a goniometer. The maximal push towards the leg stand was held for 3–5 s and performed twice with 30–60 s rest between the sets. The produced force was recorded by an isometric strain-gauge dynamometer. A minimum of two trials were performed for each subject and the best result was selected for further analysis. 

All methods (trunk flexion, extension and leg extension measurements with electromechanical dynamometer) are well documented and used in previous studies [12]. The reproducibility of measurements of maximal isometric muscle force is high (r = 0.98, C.V. =4.1%) [13]. Moreover, the device used in the study had been regularly calibrated. Finally, overall maximal muscle strength in the present study refers to the results of these three measurements (leg extension and trunk flexion and extension). 

### 2.3. Muscular Endurance Tests

A modified three stage sit-up test was used to measure muscle endurance of the core (especially abdominal and hip flexor) muscles. The three-stage sit-up test has been used with population-level studies [14]. In this test, the subject assumed a supine position on a mat with the knees at 90 degrees and feet flat on a mat where they stayed throughout the test. For the first ten repetitions, the subject placed his hands on the thighs and curled up until the hands reached the kneecaps. For the next ten repetitions (from 11 to 20), the subject had the hands across the chest and during the third set (from 21 to 30 repetitions) the subject pinched his ears. The assistant held the subject’s feet on the ground on these first three sets (repetitions from 1–30). Respectively, in the next three sets, repetitions from 31–40, 41–50 and 51–70 were done in the same way with the exception that the assistant was not holding the feet and the last set consisted of 20 instead of 10 repetitions. The subject sat up to full sitting position and then returned back to the floor (back of the head touching the floor) and continued to perform a maximum of 70 sit-ups. The test was stopped if the subject could not sit up or the feet were not touching the floor in the last 40 repetitions.

A seated alternative dumbbell press was used for measuring upper body muscle endurance. In this test, the subject held a 10 kg dumbbell in each hand at the shoulder level. The subject alternately raised the dumbbells from their shoulders with palms facing side and elbows pointing forward to full extension of elbow. Only one hand was allowed to move each time and no leaning forward or jerking the weight during the movement was allowed. 

### 2.4. Statistical Analysis

Standard statistical methods were used for the calculation of the descriptive statistics (means and standard deviations) with the analysis of normal distribution in each variable. Statistical analyses between the initial and the second test were normally distributed and done using Student´s paired t-test. The changes between baseline and follow-up tests in fitness variables were not normally distributed and therefore correlations were done using Spearman correlations. Effect sizes were assessed by calculating differences of means (post–pre) divided by baseline standard deviation (pre). Statistical significance was accepted at *p* < 0.05. SPSS statistic (version 24) (IBM Corp., Armonk, NY, USA) commercial computer program was used in statistical analysis. 

## 3. Results

The maximal isometric bilateral strength of the leg extensor muscles increased by 5% (220 ± 42 to 232 ± 42 kg, *p* < 0.01, ES 0.28), whereas trunk extension strength increased by 3% (117 ± 21 to 120 ± 19 kg, *p* = 0.03, ES 0.13) and trunk flexion by 5% (82 ± 16 to 86 ± 17 kg, *p* < 0.01, ES 0.25), respectively (Figure 1). Moreover, the isometric test results between the first and second year studies were compared as weight-adjusted (kg/body weight) mean. The differences in leg extension (from 2.9 ± 0.5 to 3.0 ± 0.5 kg/body weight, *p* < 0.01, ES 0.26) and trunk extension (from 1.5 ± 0.2 to 1.6 ± 0.2 kg/body weight, *p* < 0.01, ES 0.08) test results remained statistically significant when expressed relative to body weight, whereas trunk flexion test results (from 1.1 ± 0.2 to 1.1 ± 0.2 kg/body weight, *p* < 0.01, ES 0.24) were not significant (p 0.15). The year-by-year analysis revealed variation in isometric strength between the cadet courses. The range of the follow-up results in the leg extension was 194–268 kg and, respectively, 110–135 kg in trunk extension and 74–102 kg in trunk flexion between the years.

The upper body muscle endurance test results increased from 68 ± 13 to 75 ± 15 (rep/min) repetitions (*p* < 0.01, ES 0.40) and the modified sit-up test results from 47 ± 12 to 51 ± 13 (rep/min) repetitions (*p* < 0.01, ES 0.20), respectively (Figure 2). The range of the test results between the different cadet courses in follow-up were 62–87 rep/min in upper body muscle endurance test and 42–59 rep/min in modified sit-ups, respectively. Furthermore, the mean body mass increased from 76.4 ± 7.2 to 77.3 ± 7.2 kg (*p* < 0.01, ES 0.17) and BMI increased from 23.6 ± 1.8 to 23.9 ± 1.8 kg/m^2^ (*p* < 0.01, ES 0.15) between the initial test and the second year test. There was no change in height of the subjects between the tests. 

When the changes between the initial and follow-up tests of all results (as weight-normalized force output) were combined together, the change in the isometric leg extension test was weakly and positively correlated with the change in the isometric trunk flexion (r = 0.29, *p* < 0.01) and extension (r = 0.31, *p* < 0.01) tests. The change of isometric extension had a significant but weak positive correlation with the isometric flexion (r = 0.39, *p* < 0.01). There was no significant correlation between the changes in results of muscle endurance tests. The change in body mass correlated with the changes in results of isometric trunk flexion (r = 0.18, *p* = 0.02) and extension (r = 0.16 *p* = 0.04) and leg extension (r = 0.16, *p* < 0.04) tests.

## 4. Discussion

The results of the present study showed slight improvement in both maximal isometric strength and muscle endurance test results measured in the first and second year of studies among AFA cadets. In addition, the improvements in maximal strength tests correlated weakly and positively together. 

The results in the maximal isometric bilateral leg extension test indicates maximal strength in all thigh and gluteal muscles responsible for leg extension movement and, therefore, achieving a sufficient level in this test is important in many daily tasks in the military environment, including flying HPA. Respectively, the improvement in isometric trunk flexion and extension tests indicates maximal force in both low back and abdominal core muscles. The adequate low back strength may prevent low back disorders among military pilots and, therefore, achieving a sufficient level in this test is important for pilots flying HPA [7]. 

Even though the correlation between maximum strength and increase of Gz tolerance has not been well elucidated, a sufficient level in both isometric abdominal and leg muscle force is important for HPA pilots in the anti-G straining maneuvers (AGSM) [1]. The primary muscles activated in AGSM are particularly the gluteal and abdominal muscles [1]. The maximal strength of the gluteal muscles and the maximal strength and muscle endurance of the abdominal muscles were tested among our subjects. Even with the help of modern anti-G protection, including an anti-G suit and positive pressure breathing apparatuses, it is essential for pilots to perform adequate AGSM. The pilots exposed to G levels higher than 4–5 Gz need the protection of either suit or AGSM. Therefore, the possible failing of an anti-G suit might lead to a situation where the pilot needs to be able to produce sufficient pressure with his leg and abdominal muscles to avoid a head-level blood pressure drop. Insufficient AGSM might have fatal consequences if the pilot is not able to prevent G-induced loss of consciousness. Therefore, the improvements in muscular fitness performance during the first and second year of the AFA cadets can indicate improved occupational flight ability. 

Poor muscle fitness has been reported as a predictor and risk factor of low back pain in the general population [6] and, therefore, adequate level of muscle fitness can be presumed to be essential for military aviators flying HPA to avoid work-related musculoskeletal disorders. Better muscle fitness might not only prevent Gz-induced inflight pain episodes but it has also been associated with lower levels of sick leave among military personnel [15]. Moreover, physical activity can influence the performance of many different cognitive tasks according to a meta-analysis by Chang et al. [5]. For example, Bullock et al. [16] found that aerobic capacity is an important determinant of visual search performance under physical stress. However, the role of muscle strength and endurance in operational performance, particularly in HPA flight, has not been directly investigated. The change in physical fitness levels during basic training or early years of military academies has been studied previously [9,10,11]. However, most of the studies lack information on isometric muscle force and concentrate on measuring aerobic fitness only. We were able to find only one study that included maximal isometric tests during (U.S. Military Academy) cadet training [10]. They observed significant changes in upper torso strength in a one-year follow-up. There were no similar changes in leg extension or trunk extension test results in the one-year follow-up, which is not in line with the present study. One reason for this could be the guidance and programs tailored for pilots on PEC. Unfortunately, we did not have questionnaires, data or any kind of follow-up of the physical activity before and after the first test, nor it is possible to conduct a RCT study design to show causality. Thus, this only leads us to speculate that there might be a positive effect due to PEC on subjects’ physical fitness.

The relationships between changes in isometric strength tests were weak, but, however, were positive. These results may indicate that individuals with lower body maximal strength were also able to improve maximal strength in trunk muscles. Nevertheless, no associations were observed between changes in muscular endurance tests, indicating either that if improvements exist for muscular endurance in core muscles there are no apparent improvements in upper body muscular endurance. The reason for these non-significant associations remains unclear; however, it might be reflective of the cadets’ physical training, so that the training volume, intensity and frequency might be different for core and upper body muscular endurance. Respectively, the associations between changes in BMI or weight and isometric strength tests were weak, but positive. This indicates that individuals who improved maximal isometric strength also increased their body weight and BMI. Therefore, the increase in weight and BMI may be the result of increased muscle mass. However, we did not measure body composition or investigate the effect of diet and therefore, it is not a justified conclusion.

The strength of the present study is the use of standardized test methods with high reproducibility for measuring maximal strength for a period of ten years [13]. Further, the measurements were conducted taking into account the academic year of the subjects: the tests (baseline and control) were both organized at the same time during the mid-academic year and were not directly after the summer holidays or any excessive flight training period which could have influenced the results. Another strength of the present study is the longitudinal follow-up period (one year). According to Aandstad et al. [9], only two longitudinal studies [10,11] investigating changes in physical fitness among cadets have been conducted before their study. Most studies concentrate on very short-term follow-up periods such as a couple of months.

There are also limitations to the study. We consider that one limitation of the study is the relatively small number of subjects. Although all AFA cadets participating in PEC also participated in this study, we were not able to get more than the average of 19 cadets per year. We consider the drop-out rate (6%) very low, while only nine cadets (1–2 per year) did not participate in the second test, which highlights the reliability of our follow-up. When the results of different academic years were compared we found out that there was variation between the years in all maximal strength and muscular endurance measurements. However, our objective was to provide information on overall results during a 10 year time span, not to evaluate the variation between the different years. Another limitation of the study is the lack of information of the quantity and quality of the physical training during the follow-up. Even though we are well aware of the syllabus of the physical training at the AFA, most pilot cadets are active and participate in different sports programs during their leisure time as well. Guided mandatory physical education at the AFA is only 2 hrs/week, and, therefore, diaries or questionnaires on the amount of leisure time activity would have been beneficial. Because there was no specialized training connected to higher accelerations and g-force during the first two years of academy studies we may assume that the change in strength is the combined effect of standard physical training of cadets and leisure time activities. For future studies, we recommend longer follow-up periods to reveal the level and the change of physical fitness not just in one year, but also more importantly during the whole period of studies in AFA and during their pilot career. Moreover, we recommend examining the relationship between muscle strength and HPA flight performance in the future, rather than only acquiring more information about physical training.

## 5. Conclusions

In conclusion, the FINAF cadets’ muscular fitness improved between the first and the second year of their studies. It seems that physical education in the early phase of academy studies has positive effects on muscular fitness among pilot cadets. In addition, physical training programming should also be implemented on a regular basis through the career of a pilot.

## Figures and Tables

**Figure 1 ijerph-17-09168-f001:**
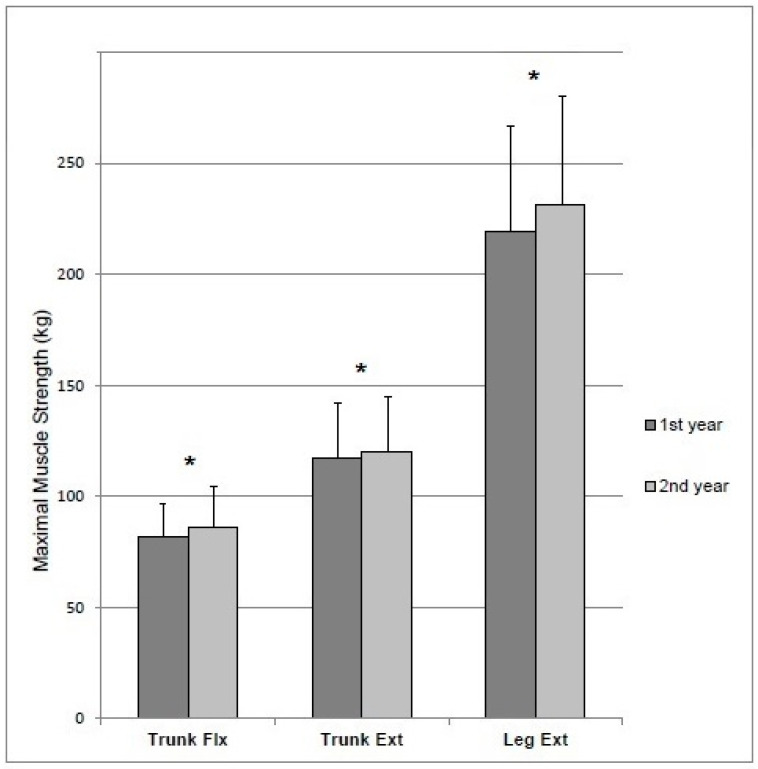
Mean (SD) maximal muscle strength of the trunk flexor, trunk extensor and leg extensor muscles during the first and second academic year. * *p* < 0.05.

**Figure 2 ijerph-17-09168-f002:**
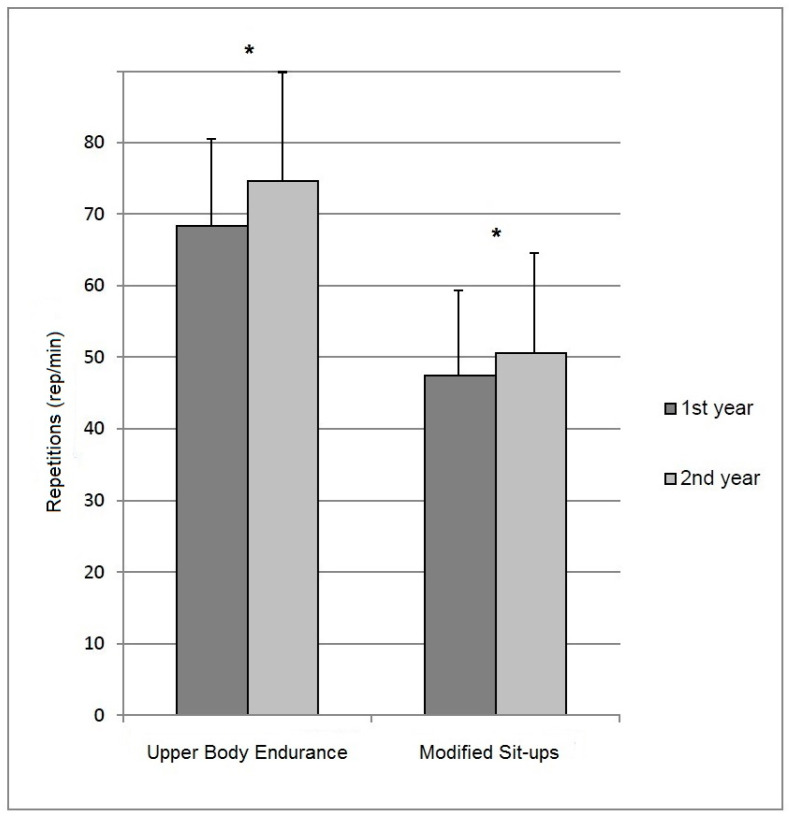
Mean (SD) maximal dynamic muscle strength of the upper limb and trunk flexor muscles during the first and second academic year. * *p* < 0.05.

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
