# Peer review of "Muscular Fitness Improves during the First Year of Academy Studies among Fighter Pilot Cadets"

_ijerph, 2020, doi:10.3390/ijerph17249168_

Round 1

Reviewer 1 Report

Thanks you for the opportunity to review this manuscript. 

There are no major concerns and I have annotated the manuscript with comments and suggestions for improvement. 

Reviewer 2 Report

Manuscript (MS) "Muscular fitness improves during the first year of academy studies among fighter pilot cadets" is connected to a potentially interesting area, unfortunately, research is not well designed. The main problem - there is not any control in the research. It implicates other problems and questions (see below).

Main remarks:

  1. The overall conception is not clear. It's hard to believe that the study design was made before research. It looks rather like a post factum calculating of existing results.

        Moreover, it’s not a longitudinal study as the Authors have stated.

  1. It’s hard to make any conclusions from the MS because of lack of the control group. One cannot be sure what was the reason of observed changes. There are some questions connected o this problem:
  • Was the change in strength the effect of standard physical training of cadets or rather an effect of specialized training connected to higher accelerations and g-force? Impossible to find without control. It’s a pity because it would be the most valuable information.
  • What was the effect of diet (connection to muscle mass)? Was it the same for all for the whole year?
  • What was a habitual activity for subjects during leisure time? Especially during holidays (Authors stated it themselves in the limitation section).
  • When was research conducted taking into account the academic year? Directly after the summer holidays? May it influence results?
  • What is known about potential changes in the physical training of cadets in the following academic years? Was this training the same?
  1. Nothing is known about variation within subgroups (for cadets of given academic year) as well as variation between subgroups (cadets from different academic years).
  2. Assuming points 1 to 3, follow up study based on different year groups without any control makes very poor scientific evidence.
  3. There is no information about effect sizes of analyzed differences (in t-Student tests d Cohen is useful). On the other hand, probably these effect sizes are very small taking into account that standard deviations in outcomes are almost three times higher than differences between means - statistical significance was probably only the effect of quite a large number of subjects.

        Also, values of R Spearman correlations are very low and one can expect  that the determination coefficient would be no higher than 10% (explained part of the whole variance).

The other remarks:

  1. The second aim is not well formulated.
  2. It’s better to use the term “subjects” than “material” when investigating people.
  3. There is no information in the Methods section about body composition measures. These were rather basic somatic characteristics then body composition.
  4. The statement in lines 85-86 is very sudden. The reader doesn’t know what is basic direction and position but is informed about the opposite direction…?
  5. Lines 90 and 92 – it’s not clear if trails and sets are the same tasks? What does it mean “minimum of two trails”? Subjects didn’t perform the same number of trials?
  6. It’s not clear why the Authors used parametric and non-parametric tests. Variables were normally distributed and differences between variables in two terms weren’t?
  7. What was the biological sense of each by each variables cross-correlation?
  8. Lines 129-130 – “Materials and Methods should be described with sufficient details to allow others to replicate and”?
  9. Lines 132-133 - “The maximal isometric bilateral strength of the leg extensor muscles increased by 5 % (220 ±42 133 to 232 ±42 kg….”. Kg is not an appropriate unit for strength, especially measured by a dynamometer. One rather expects N, kG, Dyna.
  10. Repetitions have also unit (n or n/time).
  11. Reporting results it would be better to write about “changes in results of tests” (which may indicate changes in abilities) than “changes in abilities”
  12. BMI has a unit (kg/m2).

Generally, in the reviewer’s opinion, it would be better to analyze one academic year (about 10-15 people) with a control group of similar physical training levels and show it as a quasi-experimental study with the use of two way ANOVA with “time” and “group” factors.

Reviewer 3 Report

This is an interesting paper examining the effects of training within the Finnish Air Force on muscle fitness.  A major point of novelty is the one year follow up period in a sizable group of highly specialized pilots.  However, confusion regarding methodology persists which may affect the conclusions.  Until these are clarified, it is difficult to judge the quality of this work.  Specific comments appear below.

Introduction

Lines 42-48:  The tests performed in this study do not reflect the previously identified need for low back strength and endurance.  Based on this discussion, I expect to find strength and endurance testing of the low back.  Please provide references to support the tests performed in this study.

Methodology

Line 67: units for BMI are incorrect.  Please change to m/kg2

Line 89: change “armpit” to “axillary region.” Please use similar medical anatomy language throughout.

Lines 85-99:  How are data recorded using these tests?  Does the strain gauge sample data at a given frequency and output data to an electronic device or is this an analog strain gauge with researchers recording the highest visual force output?  Was the dynamometer regularly calibrated?  Please clarify.

If using an analog strain gauge, how accurate are these measures?  This is especially important given the relatively small increase in strength observed in this study.

Did the authors measure limb length and trunk length in this study?  Given the age of participants, it is possible that some subjects grew during the year long follow up which could increase force output due to a longer lever arm rather than an increase in muscle force production.  If these measures are available, please calculate Nm of torque and report this value rather than kg of force. 

Lines 105-117:  Please write this section in past tense

Lines 118-122:  This should like a standard military press, but how are the elbows pointing forward during this test? 

Lines 129-130:  I believe this is carry-over text.  Please eliminate this sentence.

Results

The authors mention body composition data, but these do not appear in results.  Including a simple table of anthropometrics/body composition may be helpful for readers to understand the specific characteristics of these subjects.

Line 137-138:  Please clarify the “weight adjusted mean” statement. Is this kg force/body weight? If so, please include units

Figure 1.  please include SD bars and flag significant differences.  Please change this figure to reflect the weight normalized data.

Lines 150-154.  Are these correlations based on the absolute force or weight normalized force output during these tests? Please clarify.

Please rearrange this section so that all muscle strength data are discussed first (absolute and normalized).  Then, discuss muscle endurance tests.

Discussion

Line 165: Ensure “HPA” is defined earlier in this manuscript; then use the acronym exclusively

The use of terms like “adequate” and “Sufficient” are somewhat ambiguous.  How are the authors arriving at the conclusion that strength is adequate?

Although generally well-written, the discussion could use some type editing for proper English grammar.

The discussion about strength and AGSM is interesting.  Is there any evidence for the strength level needed to complete this task?  I am somewhat skeptical of how the present results relate to performance of this task.  The magnitude of increase in strength in the present study is fairly small (5%).  Is this increased strength sufficient to improve performance of this maneuver or improve safety?

Line 189-191: The authors make a big leap from the relationship between VO2max and performance to asserting that muscle endurance might be a risk factor for operational readiness.  I recommended changing this sentence to “However, the role of muscle strength and endurance to operational performance, particularly in HPA flight, has not been directly investigated.”

Line 198: Please ensure “PEC” is defined earlier in the manuscript.

Line 221-231: I would be more interested in future studies examining the relationship between muscle strength and HPA flight performance rather than the stated goal of acquiring more information about physical training.

Conclusions

Line 233: Define FINAF earlier in the manuscript

Line 236: the authors do not present any data directly addressing the value of individually tailored physical training.  This statement should be modified to better reflect this study.

Round 2

Reviewer 1 Report

Thank you for amending the manuscript.

Author Response

We wish to express our gratitude for reading the manuscript. Your input is greatly appreciated.

Reviewer 2 Report

Generally, the reviewed version of the manuscript (MS) "Muscular fitness improves during the first year of academy studies among fighter pilot cadets" looks better now.

The Authors addressed my concerns to more or less extend. 

I agree with the Authors that the results presented in the MS might be valuable.  On the other hand, besides the lack of a control group, there are at least two points where I disagree with the Authors answers.

1. I asked about variation within subgroups (for cadets of a given academic year) as well as variation between subgroups (cadets from different academic years).

The Authors responded: “We did not analyze variation within the subgroups because the aim of the present study was only to investigate the muscular fitness between the first and second year of academic studies among the FINAF fighter pilots. Our objective was to provide information of overall results during 10 yrs time span, not to evaluate the variation between the different years.”

That’s a misunderstanding – my concern was not connected to a matter of aims but to a matter of methodology. The basic problem is to find if cadets from different academic years may be mixed to one analyzed group – if the variance is homogeneous between subgroups.

Taking into account the results in the MS I suspect this variation may be quite high.

2. Unit “kg” is connected to mass – it’s out of the discussion. “Kg” might be to use carefully as a unit of indirect measure (in fact, it is similar in the case of N but it’s closer to reality by similarity to SI units). So, one cannot write that muscular strength increased from “x” kg to “y” kg – it makes no sense. It should be written that the results of “z” test changed….  and it may indicate strength change.

In fact, in some places, the Authors have changed the text to the correct form but in some places have not.

Concluding, the Authors have done some work to improve their MS but some of my concerns still exist. First is the lack of control and second is connected to the lack of information about homogeneity of variance between subgroups. In consequence, the presented results are not credible even if they are interesting and valuable to some extend.

On the other hand, it's not my intention to block this MS, so I leave the final decision to the Editors.

Author Response

Reviewer #2 Comments and suggestions:

COMMENT 1. "I asked about variation within subgroups as well as variation between subgroups. The Authors responded: “We did not analyze variation within the subgroups because the aim of the present study was only to investigate the muscular fitness between the first and second year of academic studies among the FINAF fighter pilots. Our objective was to provide information of overall results during 10 yrs time span, not to evaluate the variation between the different years.” That’s a misunderstanding – my concern was not connected to a matter of aims but to a matter of methodology. The basic problem is to find if cadets from different academic years may be mixed to one analyzed group – if the variance is homogeneous between subgroups. Taking into account the results in the MS I suspect this variation may be quite high."

RESPONSE: Thank you for your valuable comment and apologies for misunderstanding during the revision in Round 1. We have now included the analysis of cadets from different academic years to the results [Lines 144-146 and 149-151] and to the discussion [Line 242-245] of the revised manuscript. However, the variation between the subgroups was not objective of the study (as we stated before) and, therefore, we have done it only by describing the range between the different years. We hope that this will now fulfil your concerns.

COMMENT 2. "Unit “kg” is connected to mass – it’s out of the discussion. “Kg” might be to use carefully as a unit of indirect measure (in fact, it is similar in the case of N but it’s closer to reality by similarity to SI units). So, one cannot write that muscular strength increased from “x” kg to “y” kg – it makes no sense. It should be written that the results of “z” test changed…. and it may indicate strength change."

RESPONSE: It is true, that kg is connected to mass. However, it is also used as unit for maximal isometric strength in the literature. There are several studies where the strength (when tested with similar device as we have) is described as "kg". We would like to highlight that we have not presented force values (unit = N) here, which might lead to misunderstanding in the previous response (in round 1.). Further, we have modified text and deleted unnecessary and misleading word "force" in the first paragraph of the results [line 140]. Thus, we would like to thank you for your constructive criticism, time, and effort for reading this manuscript.